# A prediction method of fire frequency: Based on the optimization of SARIMA model

**Shuqi Ma**[1☯]*, **Qianyi Liu**[1☯], **Yudong Zhang**[2☯]

**1** School of Management Engineering, Capital University of Economics and Business, Beijing, China,
**2** Institute of Public Safety, Tsinghua University, Beijing, China

☯ These authors contributed equally to this work.
* shuqi_10@163.com

## Abstract

In the current study, based on the national fire statistics from 2003 to 2017, we analyzed the 24-hour occurrence regularity of fire in China to study the occurrence regularity and influencing factors of fire and provide a reference for scientific and effective fire prevention. The results show that the frequency of fire is low from 0 to 6 at night, accounting for about 13.48%, but the death toll due to fire is relatively high, accounting for about 39.90%. Considering the strong seasonal characteristics of the time series of monthly fire frequency, the SARIMA model predicts the fire frequency. According to the characteristics of time series data and prediction results, an optimized Seasonal Autoregressive Integrated Moving Average Model (SARIMA) model based on Quantile outlier detection method and similar mean interpolation method is proposed, and finally, the optimal model is constructed as SARIMA (1,1,1) (1,1,1) 12 for prediction. The results show that: according to the optimized SARIMA model to predict the number of fires in 2018 and 2019, the root mean square error of the fitting results is 2826.93, which is less than that of the SARIMA model, indicating that the improved SARIMA model has a better fitting effect. The accuracy of the results is increased by 11.5%. These findings verified that the optimized SARIMA model is an effective improvement for the series with quantile outliers, and it is more suitable for the data prediction with seasonal characteristics. The research results can better mine the law of fire aggregation and provide theoretical support for fire prevention and control work of the fire department.

## Introduction

Fire is one of the main types of disasters faced by human beings. According to the survey and statistics of international fire departments, there are about 67 million fires in the world every year, and about 60000–70000 people are lost in the fire [1]. Fire often causes great losses because of its high frequency, strong burst, great harm and difficult disposal. For example, a major fire accident occurred in Luoyang Dongdu commercial building on the evening of December 25, 2000 in China, which resulted in 309 deaths. The nightclub fire in Santa Maria city occurred in the early morning of January 27, 2013, which resulted in 242 deaths and 168 injuries.

**Data Availability Statement:** All relevant data are within the paper and its Supporting information files.

**Funding:** This work is funded by National Key R&D Program of China (No. 2018YFC0809700), the

basic research funds of Capital University of Economics and Business (No.XRZ2020018).

**Competing interests:** The authors have declared that no competing interests exist.

The huge forest fire in Australia in 2019, covering an area of 63000 square kilometers, caused an economic loss of about 5 billion Australian dollars. In recent years, the world has paid close attention to the fire problem, especially in China, great attention has been paid to the fire control work. The Fire and Rescue Department Ministry of Emergency Management has accumulated a large amount of fire data, including the time, location, casualties, economic losses, fire causes and so on, which provides a good data basis for the study of fire occurrence. In the past, descriptive statistical methods were often used in the fire situation analysis of natural fire and anthropological fire. In the past, descriptive statistical methods were often used in the fire situation analysis of natural fire and anthropological fire [2]. However, effective prediction of fire frequency in the short and medium term can provide a reliable basis for the national scientific deployment of firefighters and fire supplies, and also provide scientific theory and support for the national fire work strategic planning.

With the application of big data prediction technology in various fields, it also provides methods for the development of fire risk prediction technology. At present, there are many methods for the research of data prediction technology globally and are widely used with promising results [3, 4]. Fire risk prediction has been widely used in forest fire prevention and control. For example, Xu Aijun [5] established a forest fire prediction model by combining GIS and remote sensing technology. Yang Zi [6] established a neural network prediction model of forest fire considering many factors such as light, temperature, humidity, wind, terrain and fuel. Oulad Sayad Y [7] uses Artificial Neural Network(ANN) and Support Vector Machine(SVM) to process the data collected from satellite images to predict wildfires and avoid such disasters. The experimental results show good prediction performance. John Ray Bergado [8] use big geodata—in the form of remotely sensed images, ground-based sensor observations, and topographical datasets to design a deep fully convolutional network, called AllConvNet, to predict daily maps of a wildfire burn over the next 7 days.

With the advancement in data analysis technology, the prediction of urban fire risk has been extensively studied and applied worldwide [9]. For example, the construction of fire information in European and American started earlier than other countries, and many cities in these countries have established highly perfect urban fire alarm systems, which usually have comprehensive prevention and control capabilities such as fire data collection, summary, statistics, prediction and early warning [10]. Moreover, the fire management center can respond in time, and make analysis and decision rapidly [11]. There are many methods for fire prediction research, including time series [12], neural network prediction [13], weighted logistic regression model [14], grey system theory [15], wavelet analysis and other machine learning algorithms, which have achieved good results. For example, Zeng Mingjie [16] introduced the working principle of big data building fire risk prediction system based on machine learning and intelligent modeling. Later, the Seasonal Autoregressive Integrated Moving Average Model (SARIMA) model added the stochastic seasonal model based on constructing time series, which is more suitable for the time series model with seasonal variation characteristics [17]. The prediction of fire has been focused on the prediction of forest fire in the early stage of this century. In recent years, the fire prediction has been developed into the prediction of urban fire, mainly used to guide the fire prevention work. By analyzing the development and exploration process of fire risk prediction technology, we found that there are common problems such as model accuracy, data quality, factor selection and so on. In addition, the research results focus on the prediction of a certain type of fire or the degree of fire risk, and the analysis of the national fire development situation is limited.

In order to realize the time series prediction model suitable for the characteristics of fire occurrence, the SARIMA model is constructed and optimized according to the national fire

data and characteristics from 2003 to 2017 to provide theoretical support for fire prevention and control work of fire departments.

## Data and methods

### Data sources

The fire data is from the China Fire Services [18] from 2004 to 2018, including the monthly number of fires, deaths, injuries, causes of fire, 24-hour fire and casualties in China from 2003 to 2017.

### Model construction

Auto regressive integrated moving average (ARIMA) model is a time series prediction method proposed by Geogre Box and Gwilym Jenkins [19]. ARIMA model is a classical method of time series analysis, and is widely used. The SARIMA model is developed on the basis of ARIMA model. If the original series has obvious time fluctuation trend and seasonal characteristics, SARIMA model can be considered [20]. SARIMA model constructs a seasonal time series model by fusing ARIMA model and Stochastic Seasonal Model, which is abbreviated as SARIMA(p, d, q) (P, D, Q)S, where p and q are the order of autoregressive and moving average, P and Q are the order of seasonal autoregressive and moving average, d is the difference times, D is the seasonal difference times, S is the seasonal period and cycle length. The structure of SARIMA model is shown in Eq (1).

$$
\begin{cases}
\Phi(L)A_P(L^s)(\nabla^d \nabla_s^D x_i) = \Theta_q(L)B_q(L^s)\varepsilon_t, \\
E(\varepsilon_t) = 0, Var(\varepsilon_t) = \sigma_s^2, E(\varepsilon_t | \varepsilon_s) = 0, s \neq t, \\
\qquad E(x_s \varepsilon_t) = 0, s < t.
\end{cases}
\tag{1}
$$

Where, $L$ is delay operator, $A_P(L^s)$ is $p$-order autoregressive operator, $A_q(L^s)$ is $q$-order seasonal moving average operator, $\nabla^d = (1 - L)^d$ is difference operation, $\nabla_s^D = (1 - L^s)^d$ is seasonal difference operation.

The stationarity of time series is the basic premise of time series and processing, and non-stationary time series usually need to be converted into stationarity series. After the sequence is stabilized by seasonal and non seasonal difference, the approximate order of the model is determined according to the autocorrelation function. Then the AIC information criterion [21] is used to determine the optimal parameter combination of the model, and the model satisfies the residual white noise test.

The white noise test is a series of normal series with independent distribution, which obeys the normal distribution with mean value of 0 and variance of $\sigma^2$. Every order point of white noise obeys the normal distribution. The model of white noise is shown in Eq (2).

$$
X_t = e_t \big|_{e_t \sim WN(0, \sigma^2)}
\tag{2}
$$

Test the original hypothesis $H_0$: the residuals are white noise series; Alternative hypothesis $H_1$: the residuals are non-white noise sequences. When the original hypothesis is received, that is, the residual is white noise, all the information in the residual is mined out, and the model is good.

For evaluating the prediction model, the Mean Square Error (MSE) and Root Mean Square Error (RMSE) between the fitting value and the true value are tested. The two are performance metrics commonly used in time series analysis, and are expressed as shown in formula (3) and

formula (4):

$$MSE = \frac{1}{n}\sum_{i=1}^{n}\left(f(x_i) - y_i\right)^2 \tag{3}$$

Where, $f(x_i)$ is the predicted value, $y_i$ is the true value.

The Root Mean Square Error

$$RMSE = \sqrt{MSE} \tag{4}$$

## Results and analysis

### 24-hour variation characteristics of fire

To clarify the 24-hour change law of fire, the data of fire frequency and the death toll in China from 2015 to 2017 were collected and analyzed. The results are shown in Fig 1.

From 2015 to 2017, the fire frequency ratio 0–6 is 13.48%, while the death rate of fires at 0–6 is 39.90%. The analysis shows that this is mainly related to the certain mode of production and living habits. There are less production and life activities from 0 to 6, and the firework and fire source activity are low, so the fire frequency is low. However, people's vigilance is low in this period, the detection rate of initial fire is low, the proportion of major fires with more than 3 deaths is high, and the number of casualties is higher than that in other periods. Therefore, to strengthen the fire prevention at night and prevent the occurrence of larger fires, we should

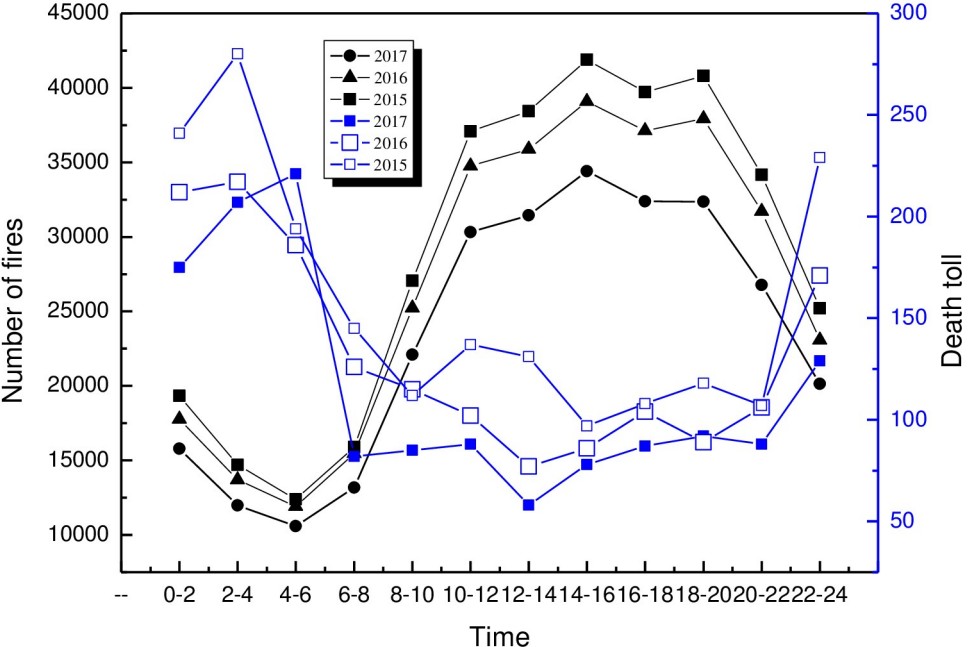

**Fig 1. Fire frequency and death toll in 24 hours in China.** From the perspective of 24-hour fire frequency in China, the fire frequency from 0 to 24 in 2015–2017 first showed decreasing from 0 to 6 and then increased slowly, and reached the peak between 14 to 20. Overall, the number of fires that occurred at 0–6 is at a low level within 24 hours. The trend of death toll is opposite to that of fire frequency. Taking 2017 as an example, a total of 38361 fires occurred from 0:00 to 6:00, with 603 deaths, with an average of 63 fires and 1 death. From 14:00 to 20:00, there were 99156 fires and 223 deaths, with an average of 444 fires and 1 death.

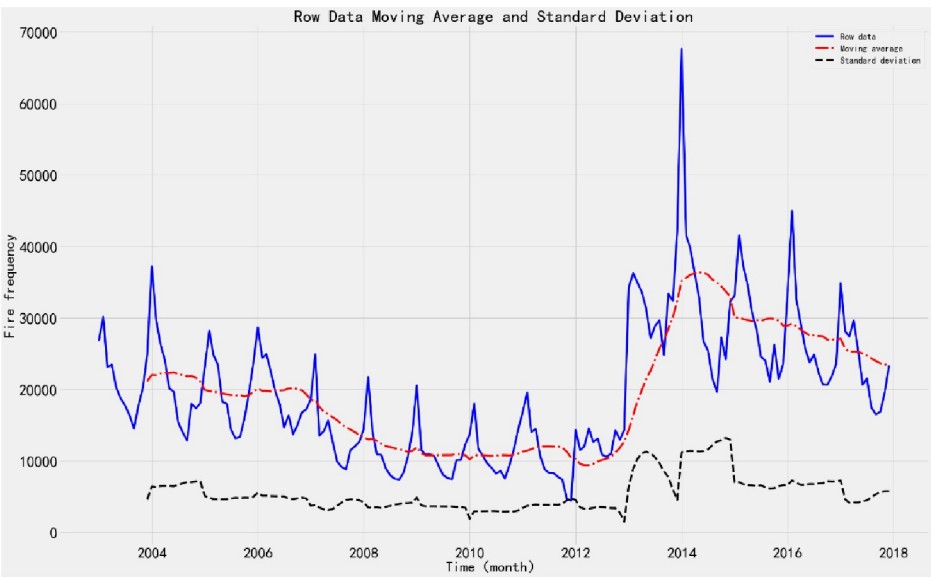

**Fig 2. ADF test after 1-order 12 step difference.** The first stage is from 2003 to 2011, showing a slow downward trend, the second stage is from 2012 to 2014, showing a rapid growth trend, and the third stage is from 2015 to 2017, showing a steady downward trend.

pay more attention to the normal use of the automatic fire alarm system at night, strengthen the safety investigation of fire doors and evacuation stairs, and the safety management of the ignition source.

## Fire characteristics and prediction

The fire data are collected from the annual fire data from 2003 to 2017. According to the time trend, it can be roughly divided into three stages, as shown in Fig 2.

**1. Monthly variation characteristics of fire.** The monthly distribution of fires in China from 2003 to 2017 is shown in Fig 2. It can be seen from the figure that the number of fires has a certain trend and seasonal effect, and the cycle length is 12 months. The main reason is that the proportion of fire caused by electrical causes is as high as 40% in the whole country. In winter and spring, people use fire, electricity and gas intensively, and the fire causing factors increase. Moreover, climate factors such as temperature, humidity and wind speed in spring have a certain correlation with the incidence of fire. In addition, according to the statistics of the fire [17], the Chinese traditional festival Spring Festival is generally from January to February. In the past 15 years, the proportion of fires during the Spring Festival (6 days from New Year's Eve to the fifth day) accounts for 30%–45% of that month's fires. There are also many fire accidents caused by fireworks during the festival.

Through the random effect of the series, it is found that the residual after the extracted trend and seasonal effect is basically stable. However, the number of fires in January 2014 is obviously abnormal, which is more than two times higher than that in the same period of each year.

**2. Fire frequency prediction.** 1. Stationarity and white noise test of time series

1) ADF test

To judge whether the time series data is stationary or not, the ADF test is performed on the time series through the TestSationaryAdfuller function in Python. Define the hypothesis

**Table 1. Test statistics of time series before and after stabilization under confidence interval.**

| Before time series stabilization | | After time series stabilization | |
|---|---|---|---|
| Test Statistic | Critical Value | Test Statistic | Critical Value |
| -3.443 | 1% | -3.474 | 1% |
| -2.867 | 5% | -2.880 | 5% |
| -2.570 | 10% | -2.577 | 10% |

$H_0$: Time series are non-stationary.

The test process is based on the comparison of test statistics and critical values under different confidence levels. If the test statistic $\chi_i$ is less than the critical value $t$, the hypothesis $H_0$ can be rejected and the sequence is considered stationary at a given significance level. Otherwise, the time series are considered to be nonstationary.

The time series of monthly fire occurrence show an obvious seasonal variation trend. ADF stationarity test is carried out on the time series, and the moving average and standard deviation are obtained, as shown in Fig 2. The value of ADF test statistic is 1.3786, and the test statistic corresponding to the confidence interval is shown in Table 1.

From the moving average and standard deviation of ADF test, we found that the change of the mean and standard deviation of fire frequency changes with time, indicating that there is a trend effect in the time series, so it can be considered that the time series is non-stationary. According to the ADF test results in Table 1, under 90%, 95% and 99% confidence levels, the values of test statistics are greater than the corresponding critical values. Therefore, there is no sufficient evidence to reject the original hypothesis and consider the time series to be non-stationary.

2) White noise test

For the original sequence, the seasonal difference order D = 1 and the seasonal period S = 12 are made. When the sequence is delayed for 1–12 periods, the p-value of Q statistic is less than 0.01. Therefore, at the significance level of 0.01, the original hypothesis is rejected; that is, the sequence after the 1-order 12- steps difference is a non-white noise sequence.

3) Time series stabilization

The difference method is used to eliminate the trend effect and seasonal effect of time series. 1-order 12-step difference is used to extract the trend effect and seasonal effect of the original sequence. The sequence diagram and test results after the difference are shown in Fig 3. According to the ADF test results after stabilization, the value of test statistics is -4.131 (Table 1). Under the 90%, 95% and 99% confidence levels, the value of test statistics is less than the corresponding critical value. Therefore, there is sufficient evidence to reject the original hypothesis that the time series after 1-order 12 step difference is stationary. According to the above analysis, the original sequence after the 1-order 12 step difference is a non-white noise stationary sequence, so the model can be established.

2. Model order determination

According to the time series, the seasonal difference order D = 1 and the non-seasonal period S = 12 are determined, and the autocorrelation (ACF) and partial autocorrelation (PACF) diagram of the difference sequence are drawn, as shown in Fig 4.

3. Ljung-Box test

Ljung-box test is used to test the model parameters. The results show that the p-value of Q statistic is greater than 0.05 when the residual sequence is delayed by 1–12 orders; therefore,

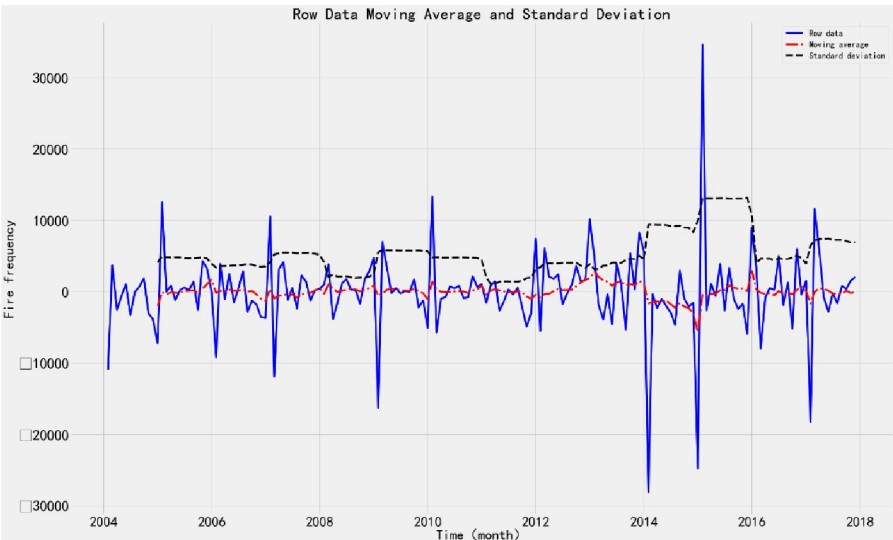

**Fig 3. ADF test after 1-order 12 step difference.** The sequence chart shows that after the difference of monthly fire quantity the moving average value fluctuates up and down in the value of 0, and the variation of variance with time is very small, indicating that the sequence after the difference is similar to the stable.

the residual is a white noise sequence at the significance level.

According to the diagnosis results of the model, the time series characteristics of the residuals are basically stable. However, there is a large fluctuation in 2014, and the distribution map is normal distribution. The results show that all parameters in SARIMA (1,1,1) (1,1,1) $_{12}$ model have statistical significance, and the model fitting effect is good.

4. Model fitting and prediction

The SARIMA (1,1,1) (1,1,1) $_{12}$ model predicts and evaluates the number of fires from January 2003 to December 2015. The mean square error (MSE) is 12852519.65, and the root mean square error (RMSE) is 3585.04. The value is small, indicating that the fitting degree of monthly fire frequency and actual monthly fire frequency is high, the overall fitting trend is consistent with the actual situation, and the effect was good.

The model is applied to predict the fire frequency from January 2016 to December 2017,

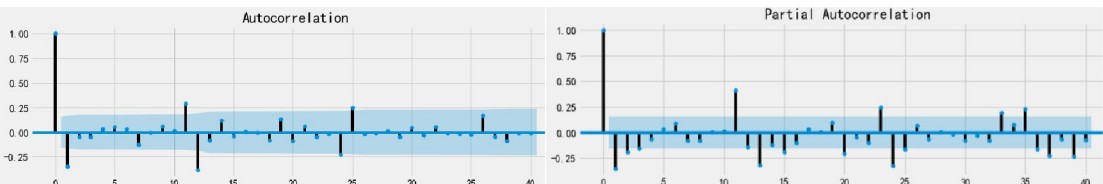

**Fig 4. ACF and PACF of the sequence after 1-order12-step difference.** The analysis of ACF and PACF shows that both the Autocorrelation Coefficient and the Partial Correlation Coefficient within the 12 steps are trailing, so p = 1 and q = 1 are used to extract the short-term autocorrelation information of the differential sequence. According to the autocorrelation and partial autocorrelation characteristics of the model, it is considered that the seasonal autocorrelation characteristics are the truncation of autocorrelation coefficient and the partial autocorrelation coefficient is tailing. Therefore, ARMA (0,1) $_{12}$ model with a 12-steps-cycle is used to extract the seasonal autocorrelation information of the difference sequence. Then the AIC criterion is used to evaluate the selected model. Through grid search and according to AIC criterion, the optimal model is SARIMA (1,1,1) (1,1,1) $_{12}$, and the AIC value is the lowest, which is 2996.15.

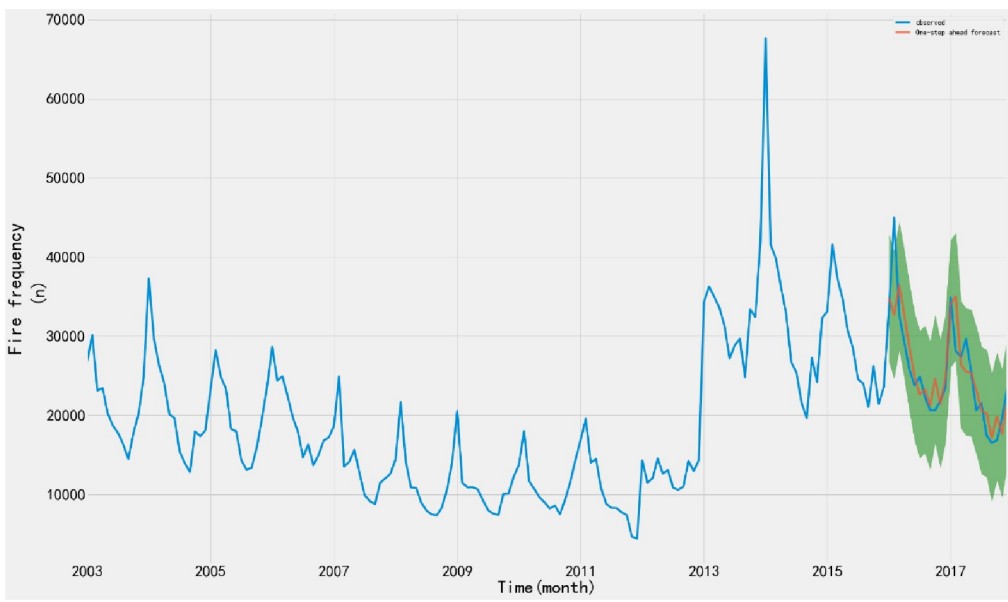

**Fig 5. Time series of fire frequency fitting.** The model is used to predict the fire data from January 2018 to December 2019, and the prediction results are shown in Fig 8. Compared with the results of the total number of fires in 2018 and 2019 published by the website of the Emergency Management Department, we found that the predicted data are larger than the actual value, and the predicted results show an upward trend, while the actual fire occurrence shows a downward trend year by year (Table 2).

and the test samples are compared with the actual data. The data of the verification set are within the 95% confidence interval of the predicted number of fires, indicating that the prediction accuracy of the model is high (Fig 5).

## Results of optimized SARIMA model and comparison

The accuracy of SARIMA model is better, but the prediction error of future data is larger than the actual value. The reason is that the number of fires in January 2014 is relatively large, which is different from that in the same period of each year. Therefore, it is speculated that there are outliers in the time series, which affect the prediction results and trends, so it is necessary to detect and process the outliers in the data series.

1. Tukey's test
   Quantile outlier detection method is used to find the maximum, minimum, median, upper and lower quartiles of the data according to the data distribution characteristics, and draw the box.

$$IQR = Q_3 - Q_1 \qquad (5)$$

**Table 2. Comparison of predicted value and actual value of SARIMA model and optimization model.**

| Year | Actual annual fire frequency | Predictive value of SARIMA model | Predictive value of optimized SARIMA model |
|------|------------------------------|----------------------------------|--------------------------------------------|
| 2018 | $2.37 \times 10^5$ | $2.89 \times 10^5$ | $2.71 \times 10^5$ |
| 2019 | $2.33 \times 10^5$ | $3.01 \times 10^5$ | $2.65 \times 10^5$ |

Where, $Q_3$ is the third quantile, $Q_1$ is the first quantile. Judge the outlier value according to Eq (6).

$$\begin{cases} x_n > Q_3 + 1.5 \times IQR \\ x_n < Q_1 - 1.5 \times IQR \end{cases} \tag{6}$$

According to the data of monthly fire frequency in China from January 2003 to December 2015, the cubic box diagram is drawn and shown in Fig 6. The results show that there are two outliers in the data sequence. It is the number of fires in January 2014 and February 2016.

2. Outlier Processing
   Outlier processing uses the same mean interpolation method for outlier interpolation and replacement. Because of the obvious periodic effect of the sequence, the mean values of the front and back in the same period are used for interpolation. The calculation method is shown in Eq (7).

$$x'_n = (x_{n-s} + x_{n+s})/2 \tag{7}$$

Where, $x'_n$ is the interpolation value, $x_{n+s}$ is the upper period value of the same type, $x_{n-s}$ is the period value of the same type, $s$ is the period of data sequence.
After calculation, the number of fires in January 2014 is interpolated as $x'_{n1} = 33729.5$, and that in February 2016 is interpolated as $x'_{n2} = 34855.5$.

3. Optimization of SARIMA model fitting and prediction results
   After Outlier processing, the optimized SARIMA $(1,1,1)$ $(1,1,1)_{12}$ model is used for fitting prediction evaluation. The overall prediction effect is good, and the prediction trend is consistent with the actual trend. The MSE is 7991582.27, and the RMSE is 2826.93, which are less than the fitting results without outlier treatment, indicating that the optimized model has a better fitting effect. The model is applied to predict the actual fire times from January 2016 to December 2017, and the test samples are compared with the actual data. The data of the verification set are within the 95% confidence interval of the predicted fire times, indicating that the prediction accuracy of the model is high (Fig 7).

## Conclusion

According to the fire frequency data in China, the SARIMA model and the optimized SARIMA model are constructed with Python for statistical analysis and prediction. The conclusions are as follows.

1. According to the 24-hour fire data, the trend is from 0 to 24, showing a trend of first decreasing and then increasing, and then again decreasing.
   The number of fires that occurred at 0–6 is less, accounting for 13.48% of the total number of fires in 24 hours, but the number of deaths due to fire is relatively high, accounting for 39.90%. From 6 to 24, there are more fires, but the proportion of casualties is low, indicating that the fire risk prevention at night should be paid more attention.

2. The fire frequency in each year has a seasonal fluctuation trend, and the number of fire occurrences in previous years has the same trend. According to this feature, SARIMA model is considered to be used for fitting and prediction. According to the fitting results of the original data series, the fitting results show that the fitting effect of the model is good,

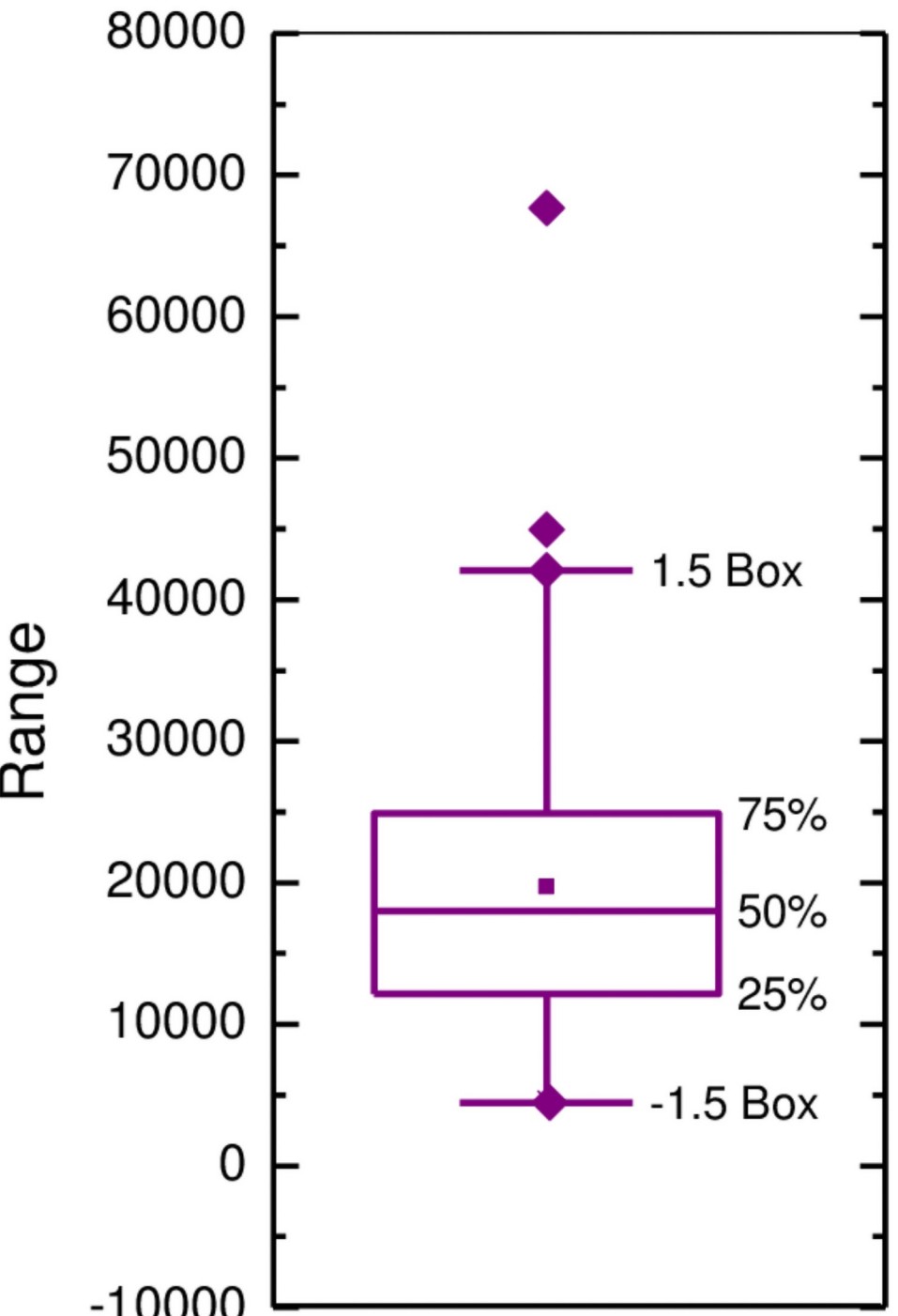

**Fig 6. The third quartile outlier detection box.** According to the monthly fire frequency data from January 2003 to December 2017, the third quartile box is shown in Fig 6. The results show that there are two outliers in the series. $x_{n1} = 67668$ is the number of fires in January 2014, and $x_{n2} = 44972$ is the number of fires in February 2016.

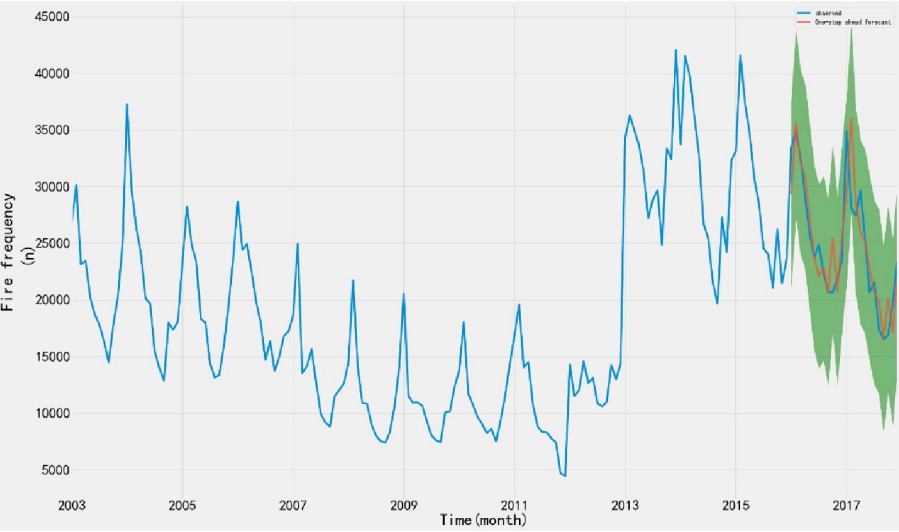

**Fig 7. Time series of fire frequency fitting by the Optimization of SARIMA model.** The optimized SARIMA model is applied to predict the fire data from January 2018 to December 2019, and the prediction results are shown in Fig 8. Compared with the prediction results of the original model, the optimized model shows that the fire frequency predicted by the optimized SARIMA model is lower than the original model. Moreover, the prediction data shows a downward trend year by year, which is consistent with the trend of the fire frequency.

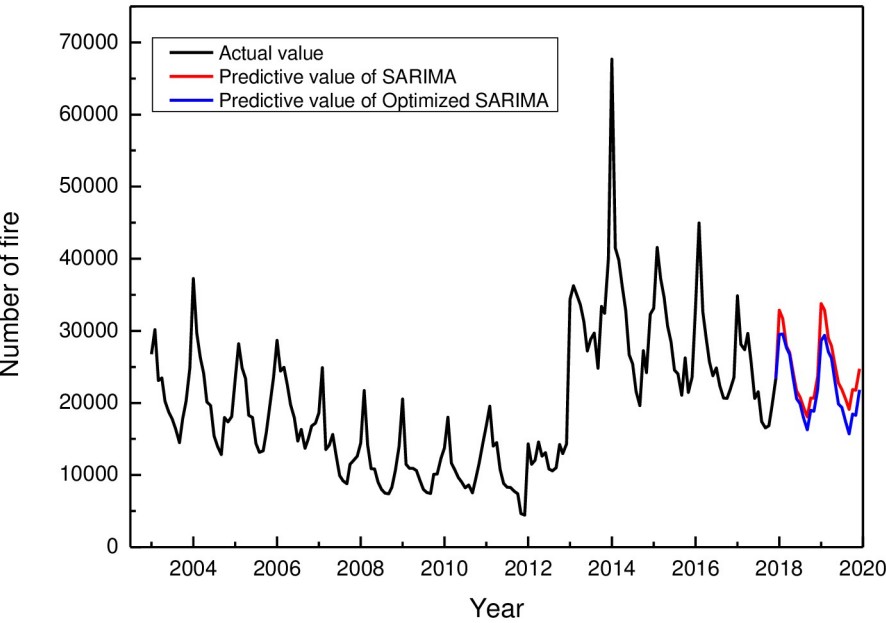

**Fig 8. The predictive value of the two models.** Compared with the total number of fires in 2018 and 2019 published on the website of the emergency management department, we found that the trend of the predicted data is consistent with the actual value, and is closer to the actual value than that of the original model. The comparison between the accumulated fire data of 2018 and 2019 obtained by the SARIMA model and optimized SARIMA model and the actual value is shown in Table 2. The accuracy of the predictive results in 2018 is improved by 7.6%, the predictive results in 2019 are improved by 15.4%, and the average accuracy is improved by 11.5%.

the MSE is 12852519.65, and the RMSE is 3585.04. The fitting trend of monthly fire frequency is consistent with the actual situation. Therefore, SARIMA model can be used to predict the number of fires.

3. The SARIMA model is optimized because the cumulative fire frequency in the next 24 months predicted by the model is higher than the actual number of fires in the year. After optimized by the method of detection of the third percentile outliers and interpolation of the same kind of mean value, the optimized SARIMA $(1,1,1)$ $(1,1,1)_{12}$ model is used for fitting prediction. The MSE is 7991582.27, and the RMSE is 2826.93, which are less than the simulation results without outlier processing, indicating that the optimized model has a better fitting effect. Compared with the original model, the predicted results are closer to the actual value, which indicates that the prediction effect is better.

## Supporting information

**S1 Data.**
(DOCX)

## Author Contributions

**Conceptualization:** Shuqi Ma.

**Data curation:** Shuqi Ma.

**Formal analysis:** Shuqi Ma.

**Funding acquisition:** Shuqi Ma.

**Investigation:** Shuqi Ma.

**Methodology:** Shuqi Ma, Qianyi Liu.

**Project administration:** Shuqi Ma.

**Resources:** Shuqi Ma.

**Software:** Shuqi Ma.

**Supervision:** Shuqi Ma.

**Validation:** Shuqi Ma.

**Visualization:** Shuqi Ma, Qianyi Liu.

**Writing – original draft:** Shuqi Ma.

**Writing – review & editing:** Yudong Zhang.

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
