## [Decision Letter · Decision Letter 0]

21 Jun 2021

PONE-D-21-17208

A prediction method of fire frequency: Based on the optimization of SARIMA model

PLOS ONE

Dear Dr. Ma,

Thank you for submitting your manuscript to PLOS ONE. After careful consideration, we feel that it has merit but does not fully meet PLOS ONE’s publication criteria as it currently stands. Therefore, we invite you to submit a revised version of the manuscript that addresses the points raised during the review process.

Please consider all the comments

We look forward to receiving your revised manuscript.

Kind regards,

Ahmed Mancy Mosa, Ph.D.

Academic Editor

PLOS ONE

Journal Requirements:

"This work is funded by National Key R&D Program of China (No.

2018YFC0809700), The basic research business expenses of CUEB (No.

XRZ2020018)"

"YES - Specify the role(s) played."

Reviewers' comments:

Reviewer's Responses to Questions

**Comments to the Author**

1. Is the manuscript technically sound, and do the data support the conclusions?

Reviewer #1: Partly

Reviewer #2: No

2. Has the statistical analysis been performed appropriately and rigorously? 

Reviewer #1: Yes

Reviewer #2: No

3. Have the authors made all data underlying the findings in their manuscript fully available?

Reviewer #1: Yes

Reviewer #2: Yes

4. Is the manuscript presented in an intelligible fashion and written in standard English?

Reviewer #1: Yes

Reviewer #2: No

5. Review Comments to the Author

Reviewer #1: 1.There is an error in Table 2 with 23,7 万， it's Chinese unit.

2.The fire data in the paper includes forest fires and urban fires, but forest fires are mainly affected by climatic factors, and urban fires are mainly affected by human factors. Whether the data application is reasonable with this model？

3.Please explain why the number of fires in January 2014 is obviously abnormal？

Reviewer #2: The paper requires a major revision. Generally the Introduction and Methods section are inadequate, which makes following the results and conclusions difficult. I provide technical difficulties with the paper below and stopped at the Results, because the manuscript requires a refocus and an attention to clarity to position the results and conclusion. For example, many conclusions are drawn within the results with real supporting evidence, it seems at times the authors were using conventional wisdom to explain the patterns presented. That is not sufficient for a scientific paper.

Introduction:

The Introduction is difficult to follow and does not set the reader up to understand the research objectives and how they address a gap in knowledge. The grammar needs to be addressed, and entire concepts like what type of fire events are even the focus of the paper (natural or anthropogenic) are entirely lost.

Line 15: What fire department?

Line 18: What kind of fire data, natural (e.g., forests, ecosystems) or anthropogenic (human infrastructure), because in the previous sentences the authors introduced natural and anthropogenic fire examples.

Line 19: fire frequency as it relates to what kind? (see previous comment

Line 33: Define “ANN” and “SVM”

Line 36: “used”; should “Big” be capitalized?

Line 43: earlier than what?

Line 44: fire alarm systems related to what? See Line 18 comment.

Line 46-48: Revise sentence structure

Line 55: edit stochastic

Line 59: in the early stage of what? Next sentence: what does “it” refer to?

Line 62: Not appropriate use of the pronoun “it”

Line 67: What is meant by “to realize”

Methods:

What is the China Fire YearBook? What types of fire data are the authors referring to? The methods section presents the statistical model forms, but does not present the actual approach to evaluating the fire data.

Line 80: “In the 20th century” is likely not needed consider removing.

Line 82: Revise for English

Line 97: This sentence does not make sense, please revise for grammar.

Line 98: The order of the model is determined by the autocorrelation function and the AIC information criterion?

Equation 2: The authors could consider removing this equation and simply reference Burnham and Anderson 2002; in fact, there is no reference to Burnham and Anderson 2002 and there should be for AIC.

Results:

Generally, it is difficult to understand and interpret the results because of the insufficient methods section.

Line 130: Do the authors mean from 0-6 on a 24hr basis? If so how does it reach a minimum at 6pm? And if it increases slowly from 6 am, then at 14:00 that would be 2pm which is before 6pm. It appears the time is incorrectly reported, please revise.

Line 127- 136: The authors should reconsider describing the trends presented in the graphs, and providing results that are not easily communicated or presented in the graphs.

Line 139: What are production and life rules, and what evidence supports this conclusion?

Line 143: What are above fires?

Line 141-148: This seems like a “common sense” conclusion but there is no evidence presented for this. I do not think the authors have spent time analyzing why the patterns they present occurred, just seems ike conventional wisdom was deployed.

Line 158-165: Again these conclusions just do not seem like they are steeped in any kind of analyses.

6. PLOS authors have the option to publish the peer review history of their article (what does this mean?). If published, this will include your full peer review and any attached files.

Reviewer #1: No

Reviewer #2: No

---

## [Author Response · Author response to Decision Letter 0]

15 Jul 2021

The content of the reply is in the file "Response to Reviewers".

---

## [Editor Report · Decision Letter 1]

26 Jul 2021

A prediction method of fire frequency: Based on the optimization of SARIMA model

PONE-D-21-17208R1

Dear Dr. Ma,

We’re pleased to inform you that your manuscript has been judged scientifically suitable for publication and will be formally accepted for publication once it meets all outstanding technical requirements.

Kind regards,

Ahmed Mancy Mosa, Ph.D.

Academic Editor

PLOS ONE
---

## [Editor Report · Acceptance letter]

28 Jul 2021

PONE-D-21-17208R1 

A prediction method of fire frequency: Based on the optimization of SARIMA model 

Dear Dr. Ma:

I'm pleased to inform you that your manuscript has been deemed suitable for publication in PLOS ONE. Congratulations! Your manuscript is now with our production department. 

Kind regards, 

on behalf of

Dr. Ahmed Mancy Mosa 

Academic Editor

PLOS ONE